# Laser Alloying Monel 400 with Amorphous Boron to Obtain Hard Coatings

**DOI:** 10.3390/ma12213494

**Published:** 2019-10-25

**Authors:** Mateusz Kuklinski, Aneta Bartkowska, Damian Przestacki

**Affiliations:** 1Faculty of Mechanical Engineering and Management, Institute of Mechanical Technology Poznan, University of Technology, Piotrowo 3, 60-965 Poznań, Poland; mateusz.kuklinski@doctorate.put.poznan.pl; 2Faculty of Mechanical Engineering and Management, Institute of Material Science and Engineering, Poznan University of Technology, Jana Pawła II 24, 60-965 Poznań, Poland; aneta.bartkowska@doctorate.put.poznan.pl

**Keywords:** Monel 400, laser, alloying, boron, microstructure, microhardness

## Abstract

In this study, Monel 400 is laser heat treated and laser alloyed with boron using diode laser to obtain adequate remelting and to improve the microhardness Single laser tracks were produced on the surface with three different laser beam scanning velocities: 5, 25, and 75 m/min. In order to enrich Monel 400 with boron surfaces were covered with initial layers of two different thicknesses before the process: 100 μm and 200 μm. In all experiments, laser beam power density was equal to 178 kW/cm^2^. Produced laser tracks were investigated in areas of microstructure, depth of remelting and microhardness. It was found that remelted zones are mainly composed of dendrites and the more boron is present in the laser track, the dendritic structure more fragmented is. Depth of remelting and microhardness depend not only on the laser beam scanning velocity but also on thickness of the initial boron layer. While microhardness of Monel 400 is equal to approximately 160 HV0.1, microhardness up to 980 HV0.1 was obtained in areas laser alloyed with boron.

## 1. Introduction

Laser modification of metals allows in obtaining surface properties of different kinds. This universality results from the fact that laser is precise, non-contact and easy to control heat source [1]. Laser remelting helps to increase corrosion resistance, surface quality, and mechanical properties of bulk materials [2,3,4,5] and to improve homogeneity and adhesion to substrate of pre-deposited coatings [6,7]. Improvements in surface properties are enhanced if additional materials are delivered to the molten pool [8,9,10].

Results of nickel superalloys laser remelting without additional elements were examined in terms of microstructural changes since the end of the 20th century [11,12,13]. Over recent years some researchers have focused on laser alloying of nickel-based alloys with ceramics to obtain hard surfaces [14]. This trend is caused by the fact that methods that successfully harden steel surfaces, like carburizing and nitriding, are inefficient for nickel alloys [15,16,17,18,19]. One of the most promising method to obtain hard coatings on nickel alloys is alloying them with boron [20,21]. Studies revealed that surface layers obtained during powder-pack or gas boriding are characterized with microhardness ranging from approximately 800 HV to 2200 HV, depending on initial composition of the treated alloy [20]. It is worth noting that boriding using a laser can obtain thicker hard surface layers in a shorter time in comparison with other methods. The thickness of laser borided layers is several times higher than of diffusion borided layers, regardless of substrate material [20,22,23,24,25,26].

One of the biggest problems connected with applications of laser alloyed materials is the high roughness of remelted surfaces obtained after the process [4,27]. Some researchers focus on this problem and propose methods to minimize this unacceptable effect during laser heat treatment [28,29]. For cylindrical surfaces, the potential alternative to grinding is turning which for hard to cut materials is even more efficient if it is performed with a laser assist [30]. Moreover, laser-assisted machining provides even smoother surface on the workpiece due to the fact that the overheating of the surface layer results in the lateral plastic flow and thus formation of surface profile without the regular feed marks [31,32,33]. Due to the fact that machining with a laser heating reduces tool wear in comparison with conventional turning, its application is also economically justified which is important for the industry [30,34].

In this study, Monel 400 is investigated in the scope of influence of laser heat treatment on its surface without and with addition of boron. This material particularly requires an improvement in hardness due to the fact that its hardness is low in comparison with other nickel-based alloys and is ranged between 100 and 250 HV [35,36]. Monel 400 is an alloy composed primarily of nickel and copper. Due to its high corrosion resistance it is mainly applied in marine engineering and chemical processing. However, low hardness disqualifies this material from being used in conditions of high wear due to erosion, cavitation or adhesive wear [35,36,37].

In order to smooth laser heat-treated or borided cylindrical surfaces of Monel 400 by turning with laser assist, laser beam scanning velocities used for the process were selected on the basis of researches on machining Monel 400 and hard-to-cut materials. Monel 400 can be successfully shaped by turning using cutting speeds ranged between 40 m/min and 100 m/min. For ceramics and metal-ceramic composites of hardness over 1000 HV cutting speeds between 10 and 100 m/min are applied in the laser-assisted machining process [30,38,39,40,41]. It was decided to carry out the laser heat treatment and laser boriding of Monel 400 using laser beam scanning velocities equal to 25 m/min and 75 m/min. Moreover, in this study low laser beam scanning velocity equal to 5 m/min is used to examine properties of a thick laser alloyed layer on the surface of Monel 400. The purpose of this research is to obtain an even remelting that undoubtedly improves the surface roughness and microhardness of the surface layer produced.

## 2. Material and Methods

### 2.1. Material

In this experiment, Monel 400 is laser heat treated and laser alloyed with boron. It is a single-phase alloy and its chemical composition is given in Table 1. Initial boron layers were prepared using a mixture of 1 g of amorphous boron and 1 mL of sodium water glass which were mixed with 0.5 mL of distilled water for obtaining appropriate consistency for applying the coating on the surface. The mixture was applied to specimens using fine hair paintbrush. Specimens were 30 mm long, 20 mm wide and 12 mm thick and two different thicknesses of initial boron layer were applied: 100 μm and 200 μm. Thickness was measured with coating thickness gauge (Elcometer company, Manchester, UK) when coatings have dried.

### 2.2. Laser Heat Treatment Process

The laser heat treatment process (LHT) was carried out with diode laser TRUMPF TruDiode 3006 (TRUMPF company, Ditzingen, Germany) which maximal power is 3 kW. To manipulate the location of the laser beam, robot KUKA KR16-2 (KUKA company, Augsburg, Germany) was used. The distance between focusing lens and focal point equals f = 200 mm. Beam diameter d = 1 mm was a distance between focusing lens and laser beam incident on the sample surface. The wavelength of the laser beam was ranged from 895 nm and 1070 nm. Laser tracks were produced separately and during each remelting laser beam was started above and turned off behind the specimen to provide constant laser beam velocity on whole length of the treated track. The scheme of the process is presented in Figure 1.

Laser tracks were produced on surfaces of dimensions 30 × 20 mm, parallelly to the shorter edge. Applied laser beam power was *P* = 1400 W and the diameter of the laser beam was *d_l_* = 1 mm which means that laser beam power density was constant and equal to *q* = 178 kW/cm^2^. Three laser beam scanning velocities were applied: 5, 25, and 75 m/min. The summary of laser heat treatment process parameters is given in Table 2. Additionally, in order to determine the time of beam impact on the material as well as laser beam fluence the following formulas were used [42]:(1)$$E_{t}=d_{l}/v_{l}$$
where
*E_t_*—interaction time of laser beam on material [s],*d_l_*—laser beam diameter [mm],*v_l_*—scanning speed [mm/s].
(2)$$F=\frac{PE_{t}}{\pi r^{2}}$$
where
*F*—laser beam fluence [J/mm^2^],*P*—laser beam power [W],*r*—radius of the laser beam [mm],*E_t_*—interaction time of laser beam on material [s].

### 2.3. Specimens Preparation

Directly after producing laser tracks specimens were cut across to obtain metallographic microsections. Surfaces with visible cross-sections of laser tracks were ground with abrasive papers of grit number ranging from 120 to 2000. After grinding, polishing was carried out for 20 min. For polishing, the deagglomerated gamma-alumina powder 0.05 μm in diameter, produced by Struers, was used. The alumina powder was mixed with demineralized water according to the manufacturer’s instructions before the process. The second stage of polishing was executed in a diamond suspension in order to reduce polishing time. The last step was etching cross-sections with Marble’s reagent for 20 s.

### 2.4. Specimens Examination

Examinations were divided into three steps: surface visual investigation, observation of microstructure, and measuring depth of remelting, and microhardness testing. For visual investigation two microscopes were used. Surfaces were observed using stereomicroscope Stereo Discovery V20 (Carl Zeiss Microscopy GmbH, Oberkochen, Germany) and laser tracks’ cross-sections were examined with light microscope Opta-Tech of series LAB40 (Opta-Tech company, Warsaw, Poland). Microhardness indentations were fabricated using Zwick 3212B Vickers tester (Carl Zeiss GmbH, Oberkochen, Germany) with load of 0.9807 N. Indentations were larger than specific areas of different phases. It was important for further treatment and applications to measure overall microhardness in different distances from the surface. Dimensions of microhardness indentations and depths of laser tracks were measured using AxioVision software (Carl Zeiss Microscopy GmbH, Oberkochen, Germany).

## 3. Results and Discussion

### 3.1. Surface Visual Investigation

Directly after laser heat treatment and laser alloying Monel 400 with boron, laser tracks were observed from above using a stereomicroscope. Figure 2, Figure 3 and Figure 4 were taken during these observations and present laser tracks produced with three different laser beam scanning velocities on Monel 400 itself and with two different boron contents.

As it can be seen in Figure 2, laser tracks obtained by remelting pure Monel 400 are characterized by constant widths along each of them. The width of the laser track produced with laser beam scanning velocity equal to 5 m/min is notably higher than of these produced using increased scanning speeds. Surfaces of laser tracks produced with laser beam scanning velocities equal to 25 m/min and 75 m/min consist of smooth center and visibly more rough sides.

Tracks laser-alloyed with boron presented in Figure 3 and Figure 4 seem to be rougher on the whole surface in comparison with those obtained by remelting Monel 400 without a boron layer. Furthermore, widths of laser tracks are heterogeneous along each of them. This is the result of small differences in surface thermal properties due to irregularities of initial boron layer. Regardless of boron content—in both areas’ laser alloyed with boron, sides of laser tracks seem to be smoother than their centers if applied laser beam scanning velocity is equal to 25 m/min or 75 m/min. This effect is opposite to observations made on laser tracks produced on pure Monel 400.

Regardless of remelting Monel 400 without or with boron, surfaces of laser tracks produced with laser beam scanning velocity equal to 5 m/min contain open pores. Also, some shallow cracks were spotted on areas alloyed with boron. However, treated specimens are planned to be machined and surface artifacts will be removed during the process. Yet, it is crucial to investigate if insides of laser tracks are free of defects.

### 3.2. Microstructure and Depth of Remelting

Microstructures of laser tracks in their cross-sections are shown in Figure 5, Figure 6 and Figure 7. Each figure contains pictures of the microstructure in two magnifications. Figure 5a–c present remelted areas produced as a result of Monel 400 laser heat treatment with three different laser beam scanning velocities. Microstructures presented in Figure 6a–c and Figure 7a–c were obtained during laser alloying Monel 400 with boron, using the aforementioned laser beam scanning velocities as well.

Significant changes are visible in Monel 400 microstructure after its remelting. It is clearly noticeable that in each specimen grain fragmentation occurs, regardless of applied laser beam scanning velocity. All laser tracks presented in Figure 5a–c is mainly composed of dendritic grains and column crystals. Their sizes and directions of growth are dependent on original microstructure and solidification speed which results from applied laser heat treatment parameters and thermal properties of the substrate. Different orientations of crystals in remelted areas result from a complicated system of thermal gradients occurring in laser track during crystallization. The majority of grains are oriented orthogonally to the boundary of the remelted zones because these were the coldest regions adjacent to the molten pool. However, also areas of different grain orientations are visible in presented cross-sections, for example those seen as equiaxed grains which crystallized along laser tracks. Areas in which grains are visible as columnar and equiaxed are marked in Figure 5a–c. Additionally, a correlation between initial position of grains boundaries in the substrate, and boundaries of areas with different orientations of crystallites in remelted zones was spotted. This correlation is particularly visible in laser tracks produced with laser beam scanning velocities 5 m/min and 25 m/min. In these cross-sections boundaries between areas of different column crystals orientations are extensions of original grain boundaries but directed toward the surface. Areas, where it is visible, are marked with arrows in Figure 5a,b. This effect virtually disappears if laser beam scanning velocity equal to 75 m/min is applied. It is related to very quick solidification–so quick that material solidifies in position of liquid metal, reflecting direction of heat convection, as it is seen in Figure 5c.

Laser tracks alloyed using boron paste of initial thickness equal to 100 μm are presented in Figure 6a–c. Grain fragmentation occurred in areas of remelted Monel 400. However, higher contrasts between bright and dark zones were spotted. It is caused by heterogeneity of chemical composition due to addition of boron and composition of borides during the laser heat treatment. During laser remelting, chemical homogeneity perishes due to intensive convection in the molten pool. Furthermore, subsequent solidification led to structure fragmentation and combined with high-temperature gradient resulted in crystallization of phases supersaturated with boron. It is worth noting that orientation of column crystals in laser tracks alloyed with boron reflects directions of heat convection already if laser beam scanning velocity equal to 25 m/min is applied for the process. This effect is visible for laser-alloyed areas with lower scanning speed than for remelted Monel 400 due to a faster heat dissipation resulting from higher thermal conductivity of boron and its compounds with nickel (values are given in Table 3).

Laser tracks produced during laser-alloying Monel 400 with boron layer 200 μm thick, presented in Figure 7a–c, seem to have a more homogenous composition than these alloyed with 100 μm boron layer. Thicker boron coating contributes to smaller amount of Monel 400 substrate in coatings than it is in other samples and in laser alloying process the final chemical composition is a result of relative proportion of the substrate and addition material. Bright areas that are mostly seen in laser tracks areas are probably hard nickel borides. Different shades visible in laser tracks represent regions of various proportions between nickel and boron. On the other hand, it can be assumed that due to various chemical composition of coatings produced with same LHT parameters, structures could reveal differences in reaction with applied reagent. Moreover, boundaries between laser-alloyed tracks and the substrate are composed of dendrites which in microscopic image seem darker, suggesting that there are more elements from the substrate in them.

It is noteworthy that cross-sections presented in Figure 7b,c contains small pores in their volumes. They are the result of the quick solidification. Since laser heat treatment was performed in air, some of it penetrated the molten pool. The time of solidification was too short to release remains of air and they consolidated in volumes of laser tracks. This effect is visible only in some cases, regardless of the value of laser beam scanning velocity used for the process.

Simultaneously with an investigation of microstructure depth of remelting was measured for each of them. Each measurement was performed after regrinding the specimen and given results are average values from 5 measurements. The results of these measurements are shown in Figure 8. As can be seen, depths of laser tracks obtained due to laser treatment and laser alloying with boron using laser beam scanning velocity equal to 5 m/min are relatively large and ranged between 310 μm and 420 μm. However, these values change irregularly. The depth of remelting Monel 400 is equal to 360 μm. The addition of 100 μm of initial boron layer results in obtaining laser track only 310 μm deep and doubling the amount of boron causes increasing of laser tracks’ depth to 420 μm which is the highest measured value. This phenomenon is the result of changes in thermal conductivity, depending on composition of the molten pool. Borides composed after remelting Monel 400 with initial layer 100 μm thick are considered to be only inclusions in a large molten volume and their existence decreases total thermal conductivity. The situation changes when the amount of boron is doubled. In this case concentration of borides in the molten pool is so high that heat is conducted mainly through them and, as it can be seen in Table 3, their thermal conductivity is higher than thermal conductivity of Monel 400.

Remelted areas obtained by laser heat treatment and laser alloying of Monel 400 with boron, using laser beam scanning velocities equal to 25 m/min and 75 m/min, have from four to six times fewer depths. Moreover, the nature of changes depending on the initial boron layer thickness is different. The depth of remelting obtained during laser heat treatment of Monel 400 with laser beam scanning velocity equal to 25 m/min is 108 μm, and if the laser beam scanning velocity is 75 m/min, this value is about 90 μm. Unlike the treatment with low laser beam scanning velocity, addition of initial boron layer 100 μm thick increases depths of laser tracks and they are equal about 110 μm. This is because obtained numbers of borides in relatively small remelted areas are enough to reach concentrations at which the heat conduction occurs through these hard particles. On the other hand, depths of laser tracks are lower if thickness of the initial boron layer is equal to 200 μm. For laser beam scanning velocities equal to 25 m/min and 75 m/min this depth is about 60 μm. This reduction is considered to be a result of insufficient heating of surfaces due to the heat insulation of Monel 400 by relatively thick boron layer, according to the high value of heat capacity of boron which is given in Table 3. This effect does not occur if laser beam scanning velocity is 5 m/min because the time of heating of unit volume is significantly longer.

### 3.3. Microhardness

Microhardness testing was performed in each of laser tracks cross-sections. For each specimen minimum, three indentations were produced in the remelted area and exactly three in the substrate. Obtained results are presented in Figure 9, Figure 10 and Figure 11 as functions of distances from the treated surface. Blue curves stand for results measured after laser heat treatment of Monel 400, red curves after laser boriding with 100 μm boron layer and black curves after laser boriding with boron layer 200 μm thick. Vertical dashed lines of relevant colors indicate average depths of each laser track.

It was found that laser heat treatment of Monel 400 itself does not affect its microhardness. Regardless of laser beam scanning velocity, in both areas: remelted zone and the substrate, results are between 130 and 180 HV0.1. However, microhardness increases due to laser alloying the surface with boron. The level of this increase is dependent on the thickness of the initial boron layer and the depth of remelting which is lower if higher laser beam scanning velocity is used. Therefore, the microhardness increases with increase of the initial boron layer thickness or laser beam scanning velocity. This phenomenon is the result of different concentrations of nickel borides with microhardness ranged from 1100 to 1800 HV [20] and eutectic phase Cu-B of microhardness between 600 and 700 HV [48] in remelted areas. Due to the fact that increasing laser beam scanning velocity lowers dimensions of remelted areas, in laser tracks produced with high scanning velocities more borides are present in a unit volume. The advantage of this effect is higher microhardness. On the other hand, high concentration of boron results in obtaining hard but simultaneously brittle coating. Brittle coatings are more susceptible to wear which can result in formation of scratches and cracks which have negative influence on working conditions of final product, like scratching machine parts. As a consequence, these changes can lead to destruction of coating or the whole part and downtimes in order to repair damages.

In Figure 9, the results of microhardness testing of specimens prepared with laser beam scanning velocity equal to 5 m/min are shown. It was observed that if the treatment is realized with this low laser beam scanning velocity, remelting Monel 400 with the initial boron layer 100 μm thick led to about 50% increase in microhardness. The average value of microhardness in obtained laser track is about 240 HV0.1. Fortunately, it is possible to produce harder surfaces using the same laser beam scanning velocity by doubling the initial boron layer. This procedure led to achieving average microhardness of remelted area equal to about 350 HV0.1 which means above two times more than of Monel 400.

Five-time increase in laser beam scanning velocity in the process of alloying Monel 400 with boron layer 100 μm thick affects its microhardness disproportionately to the large decrease of the remelted zone size. This phenomenon is noticeable by comparing graphs in Figs. 9 and 10. It was found that speeding up the laser to 25 m/min resulted in obtaining an average microhardness equal to about 300 HV0.1. This result is approximately 30% higher than obtained due to laser alloying with a 100 μm boron layer using laser beam scanning velocity equal to 5 m/min. However, unlike the laser alloying with low scanning speed, increasing the thickness of initial boron layer to 200 μm led to hardening the surface significantly. As can be seen in Figure 10, it obtained microhardness is ranged between 660 and 720 HV0.1 which is four times more than hardness of the substrate.

Results of microhardness testing in areas laser heat-treated and laser alloyed with boron using laser beam scanning velocity of 75 m/min are presented in Figure 11. As can be seen, microhardness of laser track produced with initial boron layer 100 μm thick is similar to this measured in area produced using laser beam scanning velocity equal to 25 m/min and it is ranged between 280 and 340 HV0.1. On the other hand, the highest microhardness increase of all specimens was spotted after laser alloying with initial boron layer 200 μm thick using laser beam scanning velocity equal to 75 m/min. Microhardness of this area is ranged from 820 to 980 HV0.1 and compared to the substrate is five to six times higher.

Last but not least, it was found that values of microhardness in area remelted with initial boron layer 200 μm thick using laser beam scanning velocity equal to 5 m/min is almost uniform in the whole volume. Contrary to this, microhardness in laser tracks alloyed with 200 μm initial boron layer and laser beam scanning velocities higher than 25 m/min lower deeper into specimens. This effect of microhardness decreasing into the specimen is most probably associated with changing the concentration of nickel borides throughout the coating. It can be assumed that the concentration is higher near the surface and declines in the direction of substrate material. Furthermore, this fact suggests that laser alloying Monel 400 with boron using relatively low laser beam scanning velocity lead to obtain homogenous distribution of nickel borides due to thorough mixing in the molten pool.

## 4. Conclusions

Aforementioned results of laser heat treatment and laser alloying Monel 400 with boron using diode laser led to formulate following conclusions:(1)Remelting Monel 400 with diode laser leads to obtaining a dendritic structure, in which the size of dendrites lowers with increasing laser beam scanning velocity and their orientation is perpendicular to the surface on the top of specimens. Inside laser tracks orientation of dendrites is variable and depends on substrate structure and local thermal gradients in the molten pool. The addition of boron to the molten pool results in modification of dendrites’ orientation and the more borides in remelted area, the less noticeable dendrites are.(2)The depth of remelting depends not only on the laser beam scanning velocity but also on the initial content of boron. The low concentration of borides in the molten pool decreases the total thermal conductivity of the Monel 400-boron system. However, high enough amount of boron in the molten pool increases its thermal conductivity resulting in obtaining deeper laser tracks. Depth of re-melted zone decreased from 360 μm to 310 μm due to addition of initial boron layer 100 μm thick and increased to 420 μm when the thickness of initial layer was doubled.(3)Microhardness of Monel 400 does not change after remelting it using diode laser, regardless of the laser beam scanning velocity. The addition of the initial boron layer results in increasing the microhardness and the higher boron content and the laser beam scanning velocity, the harder produced surface layer is. Thus, the highest, six-times increase in microhardness (from 160 HV0.1 to 980 HV0.1) was observed in laser track produced with initial boron layer 200 μm thick using laser beam scanning velocity equal to 75 m/min.

The results of this study will be taken into consideration in further research on laser-assisted machining of borided Monel 400 in order to obtain proper dimensions of final products.

## Figures and Tables

**Figure 1 materials-12-03494-f001:**
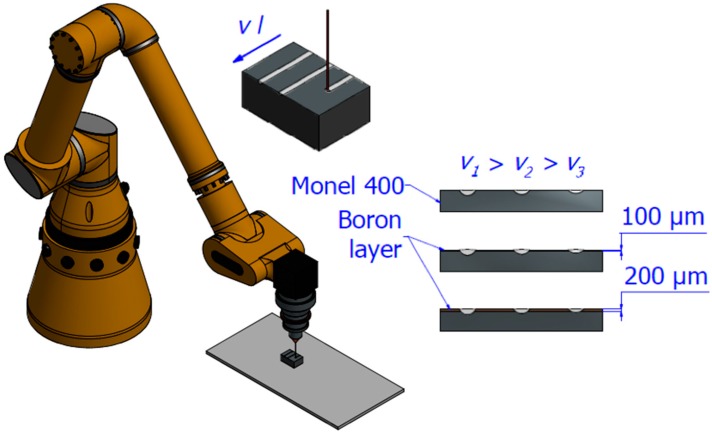
Scheme of the laser heat-treatment process.

**Figure 2 materials-12-03494-f002:**
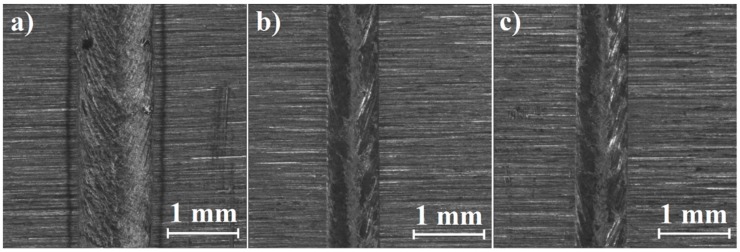
Laser tracks on Monel 400 manufactured with laser beam scanning velocity: (**a**) 5 m/min, (**b**) 25 m/min, (**c**) 75 m/min.

**Figure 3 materials-12-03494-f003:**
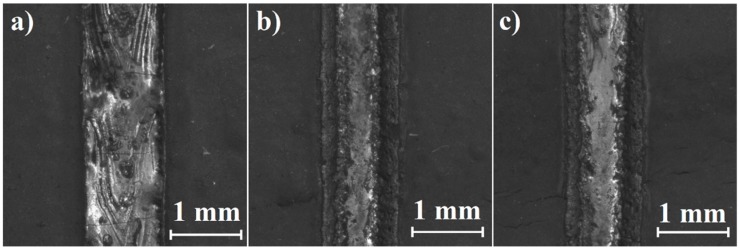
Laser tracks alloyed with 100 μm boron layer on Monel 400 manufactured with laser beam scanning velocity: (**a**) 5 m/min, (**b**) 25 m/min, (**c**) 75 m/min.

**Figure 4 materials-12-03494-f004:**
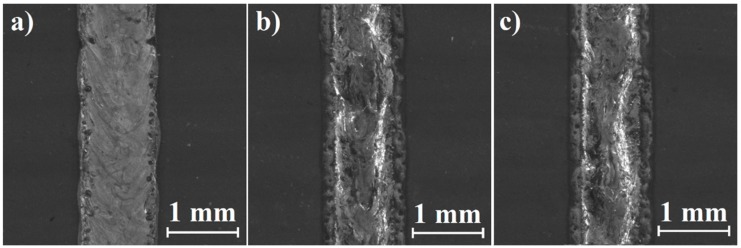
Laser tracks alloyed with a 200 μm boron layer on Monel 400 manufactured with laser beam scanning velocity: (**a**) 5 m/min, (**b**) 25 m/min, (**c**) 75 m/min.

**Figure 5 materials-12-03494-f005:**
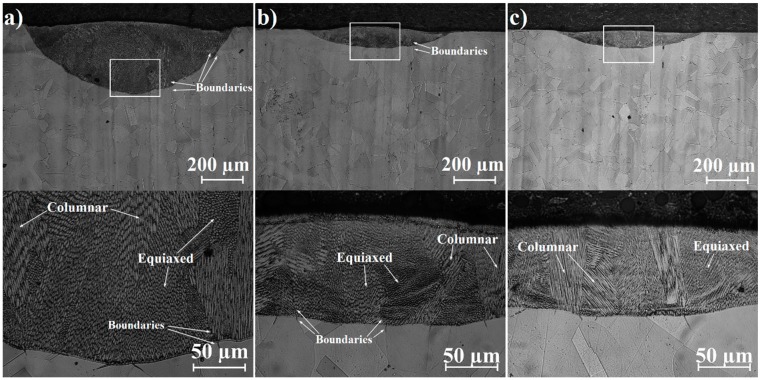
Cross-sections of laser tracks on Monel 400 manufactured with laser beam scanning velocity: (**a**) 5 m/min, (**b**) 25 m/min, (**c**) 75 m/min.

**Figure 6 materials-12-03494-f006:**
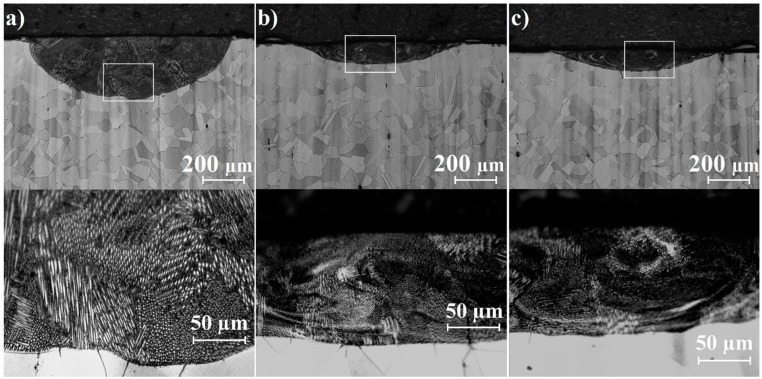
Cross-sections of laser tracks alloyed with 100 μm boron layer on Monel 400 manufactured with laser beam scanning velocity: (**a**) 5 m/min, (**b**) 25 m/min, (**c**) 75 m/min.

**Figure 7 materials-12-03494-f007:**
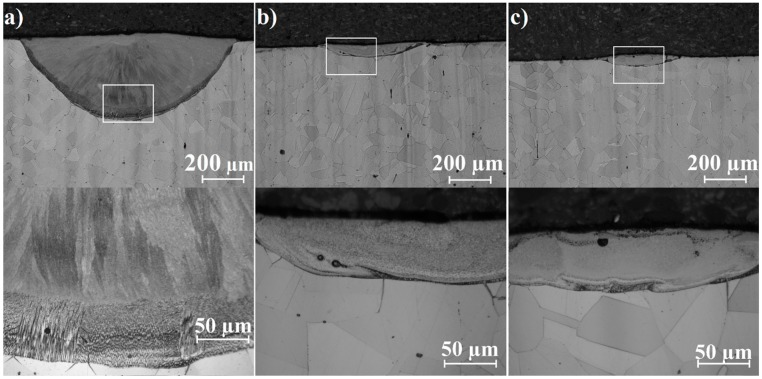
Cross-sections of laser tracks alloyed with 200 μm boron layer on Monel 400 manufactured with laser beam scanning velocity: (**a**) 5 m/min, (**b**) 25 m/min, (**c**) 75 m/min.

**Figure 8 materials-12-03494-f008:**
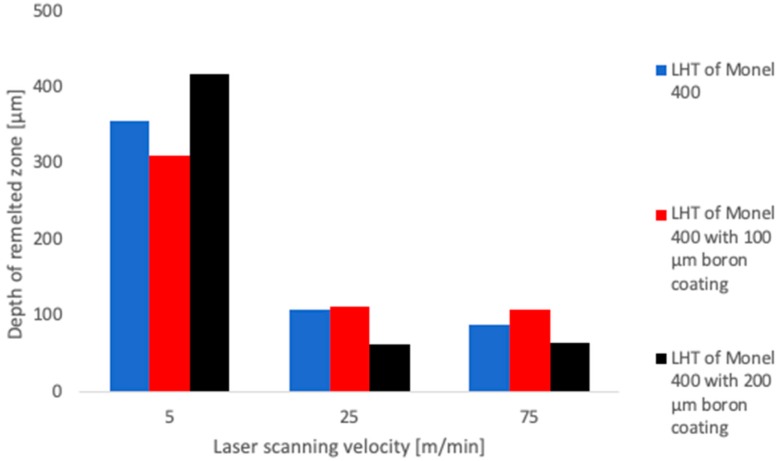
Depths of obtained laser tracks.

**Figure 9 materials-12-03494-f009:**
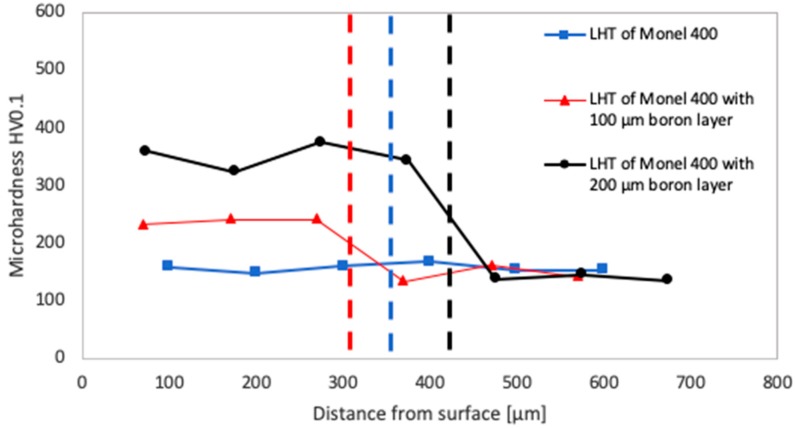
Microhardness of laser tracks manufactured with laser beam scanning velocity 5 m/min in different distances from surfaces.

**Figure 10 materials-12-03494-f010:**
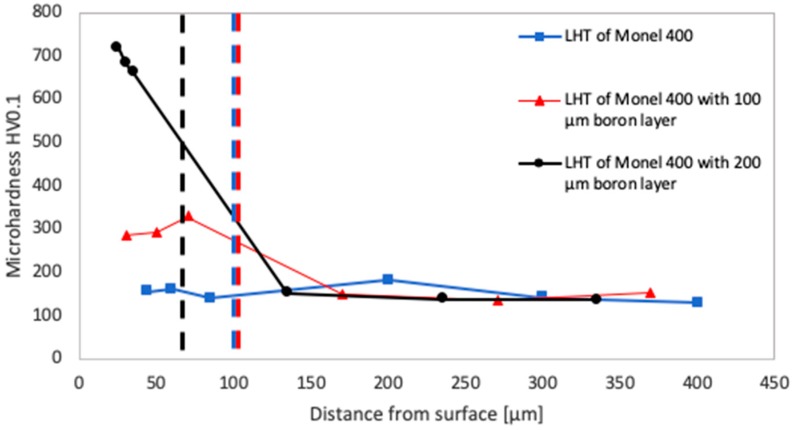
Microhardness of laser tracks manufactured with laser beam scanning velocity 25 m/min in different distances from surfaces.

**Figure 11 materials-12-03494-f011:**
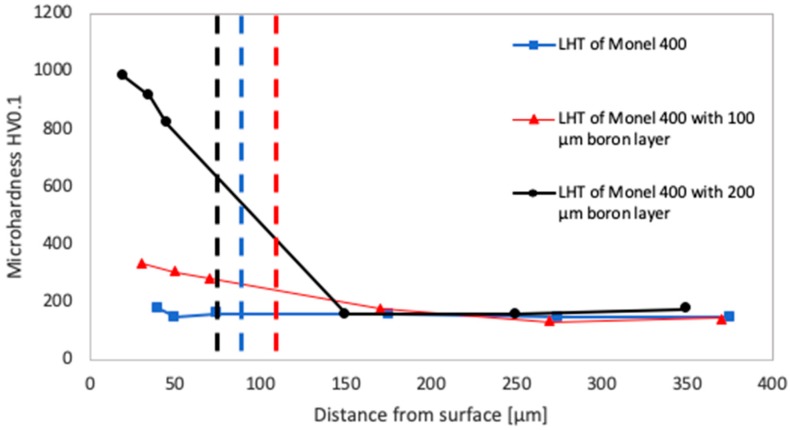
Microhardness of laser tracks manufactured with laser beam scanning velocity 75 m/min in different distances from surfaces.

**Table 1 materials-12-03494-t001:** Chemical composition of Monel 400 (wt %).

Cu	Si	Fe	Mn	C	S	Ni
31	0.5	2.5	2.0	0.3	0.024	bal.

**Table 2 materials-12-03494-t002:** Parameters of the laser heat treatment process.

*P* [W]	*q* [kW/cm^2^]	*v_l_* [m/min]	*d_l_* [mm]	*E_t_* [s]	*F* [J/mm^2^]
1400	178	5	1	0.012	21.4
		25		0.0024	4.3
		75		0.0008	1.4

**Table 3 materials-12-03494-t003:** Thermal properties of selected substances at room temperature.

Substance	Thermal Conductivity [W/(m·K)]	Heat Capacity [J/(g·K)]
Copper	400 [43]	0.384 [43]
Nickel	91 [44]	0.445 [44]
Monel 400	22 [36]	0.427 [36]
Boron	27 [45]	1.030 [45]
Ni_3_B	42 [46]	0.400 [47]
NiB	22 [47]	0.481 [48]

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
