# Peer review of "Laser Alloying Monel 400 with Amorphous Boron to Obtain Hard Coatings"

_materials, 2019, doi:10.3390/ma12213494_

Round 1
Reviewer 1 Report
This manuscript describes that Monel 400 is alloyed with boron using diode laser to be remelting while maintaining relatively high microhardness. The authors investigated the microstructure, depth of remelting and microhardness of the alloys. The topic addressed is interesting and deserves a constructive discussion. However, a revision of the manuscript is needed before it can be accepted for publication.
1. The authors are recommended to add the results of the structure of the alloys characterized by TEM, XRD etc. and those of the chemical bonding/chemical composition of the alloys by EDX, XPS etc. The findings should be discussed using the experimental data.
Author Response
Dear Reviewer.
We are very grateful for all remarks to our research paper. For us they are not only issues to edit in current paper but also clues for further publications. We hope that our answers and revisions made in the manuscript meet your expectations of the final form of our article.
Comment
The authors are recommended to add the results of the structure of the alloys characterized by TEM, XRD etc. and those of the chemical bonding/chemical composition of the alloys by EDX, XPS etc. The findings should be discussed using the experimental data.
Answer
Dear Reviewer, thank you for this remark. We are aware that results obtained by these methods are helpful. However, results given in manuscript describe some experiments which is the part of larger scientific project and at this stage specimens were not analyzed in terms of chemical bonding and composition. Unfortunately, due to the fact that we are not able to carry out suggested experiments at the moment, because of current unavailability of equipment, they will be performed in the nearest future.
Reviewer 2 Report
This study has mainly explored the effect of laser heat treatment and laser alloying coatings with boron on microstructure and surface hardness of Monel 400 material. The key point in this study is understandable, and the testing results should be valuable in some specific applications. Overall, in my opinion, the article in the current situation should be suitable for the publication in journal of materials.
Author Response
Dear Reviewer, thank you for appreciating the value and the scientific input of our research.
Reviewer 3 Report
In this manuscript, Kukliński et al. introduce the results of a morphological, structural and mechanical characterization of different samples of Monel 400 alloyed with amorphous boron. Alloying is performed by pre-deposition of a boron layer followed by laser heating, which induce the formation of a remelted material with improved hardness performance. Results presented are encouraging (hardness increases six-fold in some cases), therefore the paper is a good candidate for publication in Materials journal.
However, it must be necessarily improved. There are some errors and inaccuracies in the description of the methodology used, as well as here and there in the manuscript. Also, some points in the discussion section need to be clarified, in order not to affect the scientific soundness of the paper. Finally, I strongly recommend to improve the English used, because both language and style require extensive editing before possible publication.
In conclusion, the paper shows interesting results, but should be reconsidered after major revision.
Herewith a detailed list of comments and suggestions:
Line 11. Authors write that laser treatment (heating and alloying with boron) is performed “to obtaining adequate remelting while maintaining relatively high microhardness”. First, English should be revised (“to obtain” instead of “to obtaining”). Then, in my opinion, it should be clearly highlighted in the first two lines of the abstract that laser treatment is performed to improve microhardness, as stated in the title. It’s not clear from these first two lines, whereas it should be emphasized immediately.
Lines 19-21. Results reported (“The thickest laser alloyed zone is 420 μm and if the laser beam scanning velocity is more or equal to 25 m/min, depth of remelting is ranged between 60 and 110μm.”) cannot be appreciated without a clear description of the materials used (size, thickness of the Monel samples, etc.). I would remove them from the abstract, only highlighting the good results obtained on microhardness.
Line 21 and following . Unit of measurement of hardness in Vickers scale is not correctly expressed. Along with the hardness value expressed in HV, it should be also always reported the load in kg (as authors correctly do in some cases, e.g. at line 249 and in Figure 9). Authors are invited to check throughout the text.
Lines 22-24. Future research should never be included in the abstract, but only in the concluding remarks.
Line 31. Replace “bonding with” with “adhesion to”.
Line 71. I would replace “affects” with “improves”, which is the main goal of the authors’ experiments.
Line 77. Some more details on the coatings should be added. What is the mixing ratio of boron and water glass? And the dilution ratio of the mixture with distilled water? How have the fabricated boron coatings been applied on the Monel samples?
Line 83 and following. Some details on the laser system used are missing. Laser wavelength? Focusing system (lens, objective, magnification)? Authors write about a beam diameter: is it the beam waist on the focal plane? Please specify.
Lines 85-87. Check English used for the description of the laser movement. It doesn’t sound clear.
Table 2. On the basis of the definition of Et , the reported values of Et in Table 2 are all wrong. Let us take, for instance, the velocity vl = 5 m/min, which equals to vl = 5000(mm)/60(s) = 83.333 mm/s. Therefore, if Et = dl/vl, as indicated in Eq. (1), we have Et = 1(mm)/83.333(mm/s)=0.012 s, and not 0.72 s as reported in Table 2. The correct values of Et for the other two velocities are 0.0024 s for vl = 25 m/min and 0.0008 s for vl = 75 m/min. Of course, also the fluence values F reported in Table 2, calculated on the basis of wrong time values, are wrong as well, and should be corrected. Moreover, authors mention a reference (#42) to the equations they used for the calculation of the interaction time and beam fluence, but such equations are not reported in Ref. #42. Please check.
Line 111. Replace “grit” with “grit number” and “ranged” with “ranging”. Also, give some details about the polishing procedure of the obtained microsections.
Lines 146-149. Authors state that “sides of laser tracks seem to be shallower than their centers if applied laser beam scanning velocity is equal to 25 m/min or 75 m/min”. Frankly speaking, I cannot see how this can be inferred from a surface visual inspection. Also, they state that “This effect is opposite to observations made on laser tracks produced on pure Monel 400”. I have a suspicion; probably authors do not want to mean that sides are “shallower”, but “SMOOTHER” than centers in the boron-alloyed case, which is compatible with the observation made at lines 141-142 on pure Monel tracks (with smooth centers and rough sides). Am I right?
Figures 5, 6 and 7. In my opinion, zoomed pictures (with a higher magnification) are more significant. Therefore images should be swapped (i.e. put in the inset the lower magnification ones), or at least insets should be enlarged enough to appreciate dendritic grains and columnar crystals where present.
Line 177. Replace “oriented in the direction from the boundary of remelted areas to the surface” with “oriented orthogonally to the boundary of the remelted zones”.
Line 179. Authors write that “areas of different grain orientations are visible in presented cross-sections, for example these seen as equiaxed which crystallized along laser tracks”. Replace “these” with “those” and add “grains” to “equiaxed, otherwise the statement is not clear. Also, as a useful guide to the eye, could authors indicate with circles (or arrows) some examples of equiaxed grains and of columnar structures in their pictures? It could be very useful to the reader, in my opinion.
Lines 184-186. Referring to Figs. 5a and 5b, author state that “In these cross sections boundaries between areas of different column crystals orientations are extensions of original grain boundaries but directed toward the surface”. As in the case of equiaxed grains, authors are invited to indicate with a circle (or an arrow) in the picture an example of this correlation between the orientation of the columns and the original grain boundaries, because it’s hard to be seen, at least by me.
Lines 198-204. In this section, cross-sections of laser-alloyed samples with 200 μm boron layer are discussed. As far as I can see from figures 7a-c, pictures are very different from those taken from laser-alloyed samples with 100 μm boron layer, as well as from those taken from the pure Monel samples, which conversely are very similar. Authors are invited to elaborate a little bit more on this, highlighting that the material appears to be almost completely amorphous away from the boundary of remelted zones, which is probably the first and most obvious thing to notice when comparing Figs. 7a-c to both Figs 6a-c and 5a-c. Is it probably due to a higher content of amorphous boron in the laser-treated zone away from the boundary?
Line 200. Authors mention “bright areas”, stating that they are probably due to “hard nickel boride” inclusions. Bright zones present in 6a-c, conversely, have been generically attributed to not well-defined “borides”. Authors should be more consistent in the attribution of these bright zones in the two cases: if they precisely attribute them to nickel boride in Figs. 7a-c, why not in the case of Figs. 6a-c?
Lines 205-210. It’s not clear if pores are present in Fig.6, in Fig. 7, or both. Please be more specific.
Lines 210 and following. Authors discuss about the depth of remelted zones, providing some values. Authors should specify if such values are the result of an average calculation on the basis of a certain number of measurements performed on different tracks produced with the same LHT parameters on the same sample or not. Indeed, conclusions drawn may not be well-supported by the results if depth values have been obtained from one-shot measurements on a single track.
Section 3.3. Microhardness results are fairly presented and discussed. However, authors should clarify one point. At line 258, they state that “too high boride concentration makes alloyed area too brittle for potential applications”. Authors are invited to elaborate a little bit more on this, because it’s of interest to the discussion.
Line 298. Authors are invited to add another significant result, i.e. that the concentration of nickel borides and/or Cu-B eutectic phases, which demonstrate to increase microhardness, is maximum on the surface of the samples with 200 μm boron layer laser-alloyed at velocities higher than 25 m/min, and gradually decreases going deeper into the laser remelted zone.
Line 303. Authors state that “remelting Monel 400 with diode laser (by the way, it’s laser diode, and not diode laser) leads to obtain dendritic structure in which orientation is perpendicular to the surface”. However, on the basis of the discussion of results at lines 170-187, this is only partially true. Authors indeed write (line 175) “Different orientations in remelted areas result from complicated system of thermal gradients occurring in laser track during crystallization”. Authors are invited to write conclusions more in line with the discussion of results.
Author Response
We are very grateful for all remarks to our research paper. For us they are not only issues to edit in current paper but also clues for further publications. We hope that our answers and revisions made in the manuscript meet your expectations of the final form of our article.
All remarks given by Reviewers were taken into consideration in preparing reviewed version of our manuscript. Language mistakes, as well as inaccuracies in text were corrected. Calculation errors which were rightly checked by Reviewer 3 were also changed. Owing to comments about necessity to give more details about observed phenomena the manuscript describes them more clearly and contains more important information. Moreover, figures of obtained microstructures are now more meaningful for the reader.
Dear Reviewer, thank you for dedicating your time for revising our manuscript so detailed. We revised our manuscript in accordance with your suggestions. Our answers to your remarks, which we tried to formulate as precisely as possible, are given below.
Comment 1
Line 11. Authors write that laser treatment (heating and alloying with boron) is performed “to obtaining adequate remelting while maintaining relatively high microhardness”. First, English should be revised (“to obtain” instead of “to obtaining”). Then, in my opinion, it should be clearly highlighted in the first two lines of the abstract that laser treatment is performed to improve microhardness, as stated in the title. It’s not clear from these first two lines, whereas it should be emphasized immediately.
Answer 1
Thank you for this remark. According to the suggestion, revisions have been made in the manuscript.
Comment 2
Lines 19-21. Results reported (“The thickest laser alloyed zone is 420 μm and if the laser beam scanning velocity is more or equal to 25 m/min, depth of remelting is ranged between 60 and 110μm.”) cannot be appreciated without a clear description of the materials used (size, thickness of the Monel samples, etc.). I would remove them from the abstract, only highlighting the good results obtained on microhardness.
Answer 2
Thank you for this remark. According to the suggestion, revisions have been made in the manuscript.
Comment 3
Line 21 and following . Unit of measurement of hardness in Vickers scale is not correctly expressed. Along with the hardness value expressed in HV, it should be also always reported the load in kg (as authors correctly do in some cases, e.g. at line 249 and in Figure 9). Authors are invited to check throughout the text.
Answer 3
Thank you for this remark. According to the suggestion, revisions have been made in the manuscript.
Comment 4
Lines 22-24. Future research should never be included in the abstract, but only in the concluding remarks.
Answer 3
Thank you for this remark. According to the suggestion, information about future research was moved to conclusions.
Comment 5
Line 31. Replace “bonding with” with “adhesion to”.
Answer 5
Thank you for this remark. According to the suggestion, revision has been made in the manuscript.
Comment 6
Line 71. I would replace “affects” with “improves”, which is the main goal of the authors’ experiments.
Answer 6
Thank you for this remark. According to the suggestion, revision has been made in the manuscript.
Comment 7
Line 77. Some more details on the coatings should be added. What is the mixing ratio of boron and water glass? And the dilution ratio of the mixture with distilled water? How have the fabricated boron coatings been applied on the Monel samples?
Answer 7
Thank you for this remark. According to the suggestion, details about the initial coatings and preparing specimens for laser heat treatment have been added to the manuscript.
Comment 8
Line 83 and following. Some details on the laser system used are missing. Laser wavelength? Focusing system (lens, objective, magnification)? Authors write about a beam diameter: is it the beam waist on the focal plane? Please specify.
Answer 8
Thank you for this remark. According to the suggestion, details about the laser system have been added to the manuscript.
Comment 9
Lines 85-87. Check English used for the description of the laser movement. It doesn’t sound clear.
Answer 9
Thank you for this remark. According to the suggestion, language for the description of the laser movement was revised in the manuscript.
Comment 10
Table 2. On the basis of the definition of Et , the reported values of Et in Table 2 are all wrong. Let us take, for instance, the velocity vl = 5 m/min, which equals to vl = 5000(mm)/60(s) = 83.333 mm/s. Therefore, if Et = dl/vl, as indicated in Eq. (1), we have Et = 1(mm)/83.333(mm/s)=0.012 s, and not 0.72 s as reported in Table 2. The correct values of Et for the other two velocities are 0.0024 s for vl = 25 m/min and 0.0008 s for vl = 75 m/min. Of course, also the fluence values F reported in Table 2, calculated on the basis of wrong time values, are wrong as well, and should be corrected. Moreover, authors mention a reference (#42) to the equations they used for the calculation of the interaction time and beam fluence, but such equations are not reported in Ref. #42. Please check.
Answer 10
Thank you for this highly valuable remark. We checked results given in Table 2 and they were indeed wrongly calculated. Corrected values have been put in the manuscript. However, we checked and are sure that equations which we used for calculations are reported in Ref. #42, so we did not take any action according to this remark.
Comment 11
Line 111. Replace “grit” with “grit number” and “ranged” with “ranging”. Also, give some details about the polishing procedure of the obtained microsections.
Answer 11
Thank you for this remark. According to the suggestion, revisions have been made in the manuscript with additional details about polishing procedures in “Specimens preparation” chapter.
Comment 12
Lines 146-149. Authors state that “sides of laser tracks seem to be shallower than their centers if applied laser beam scanning velocity is equal to 25 m/min or 75 m/min”. Frankly speaking, I cannot see how this can be inferred from a surface visual inspection. Also, they state that “This effect is opposite to observations made on laser tracks produced on pure Monel 400”. I have a suspicion; probably authors do not want to mean that sides are “shallower”, but “SMOOTHER” than centers in the boron-alloyed case, which is compatible with the observation made at lines 141-142 on pure Monel tracks (with smooth centers and rough sides). Am I right?
Answer 12
Thank you for this remark. It is right – wrong word was used. Revision has been made in the manuscript.
Comment 13
Figures 5, 6 and 7. In my opinion, zoomed pictures (with a higher magnification) are more significant. Therefore images should be swapped (i.e. put in the inset the lower magnification ones), or at least insets should be enlarged enough to appreciate dendritic grains and columnar crystals where present.
Answer 13
Thank you for this remark. Zoomed pictures are now given in the manuscript with the same dimensions as these with lower magnification.
Comment 14
Line 177. Replace “oriented in the direction from the boundary of remelted areas to the surface” with “oriented orthogonally to the boundary of the remelted zones”.
Answer 14
Thank you for this remark. According to the suggestion, revision has been made in the manuscript.
Comment 15
Line 179. Authors write that “areas of different grain orientations are visible in presented cross-sections, for example these seen as equiaxed which crystallized along laser tracks”. Replace “these” with “those” and add “grains” to “equiaxed, otherwise the statement is not clear. Also, as a useful guide to the eye, could authors indicate with circles (or arrows) some examples of equiaxed grains and of columnar structures in their pictures? It could be very useful to the reader, in my opinion.
Answer 15
Thank you for this remark. According to the suggestion, revisions have been made in the manuscript and in new pictures – areas where columnar and equiaxed crystals occur are marked with arrows and titled.
Comment 16
Lines 184-186. Referring to Figs. 5a and 5b, author state that “In these cross sections boundaries between areas of different column crystals orientations are extensions of original grain boundaries but directed toward the surface”. As in the case of equiaxed grains, authors are invited to indicate with a circle (or an arrow) in the picture an example of this correlation between the orientation of the columns and the original grain boundaries, because it’s hard to be seen, at least by me.
Answer 16
Thank you for this remark. According to the suggestion, revision has been made in pictures. Grain boundaries and boundaries between areas of different orientations are marked with arrows and titled.
Comment 17
Lines 198-204. In this section, cross-sections of laser-alloyed samples with 200 μm boron layer are discussed. As far as I can see from figures 7a-c, pictures are very different from those taken from laser-alloyed samples with 100 μm boron layer, as well as from those taken from the pure Monel samples, which conversely are very similar. Authors are invited to elaborate a little bit more on this, highlighting that the material appears to be almost completely amorphous away from the boundary of remelted zones, which is probably the first and most obvious thing to notice when comparing Figs. 7a-c to both Figs 6a-c and 5a-c. Is it probably due to a higher content of amorphous boron in the laser-treated zone away from the boundary?
Answer 17
Thank you for this remark. According to the suggestion, more detailed description has been given in the manuscript.
Comment 18
Line 200. Authors mention “bright areas”, stating that they are probably due to “hard nickel boride” inclusions. Bright zones present in 6a-c, conversely, have been generically attributed to not well-defined “borides”. Authors should be more consistent in the attribution of these bright zones in the two cases: if they precisely attribute them to nickel boride in Figs. 7a-c, why not in the case of Figs. 6a-c?
Answer 18
Thank you for this remark. According to the suggestion, more detailed description has been given in the manuscript.
Comment 19
Lines 205-210. It’s not clear if pores are present in Fig.6, in Fig. 7, or both. Please be more specific.
Answer 19
Thank you for this remark. According to the suggestion, revision has been made in the manuscript.
Comment 20
Lines 210 and following. Authors discuss about the depth of remelted zones, providing some values. Authors should specify if such values are the result of an average calculation on the basis of a certain number of measurements performed on different tracks produced with the same LHT parameters on the same sample or not. Indeed, conclusions drawn may not be well-supported by the results if depth values have been obtained from one-shot measurements on a single track.
Answer 20
Thank you for this remark. It is rightly pointed out that we did not give an information about methodology of measuring depths of laser tracks. Additional information has been added to the manuscript.
Comment 21
Section 3.3. Microhardness results are fairly presented and discussed. However, authors should clarify one point. At line 258, they state that “too high boride concentration makes alloyed area too brittle for potential applications”. Authors are invited to elaborate a little bit more on this, because it’s of interest to the discussion.
Answer 21
Thank you for this remark. Additional information has been added to the manuscript.
Comment 22
Line 298. Authors are invited to add another significant result, i.e. that the concentration of nickel borides and/or Cu-B eutectic phases, which demonstrate to increase microhardness, is maximum on the surface of the samples with 200 μm boron layer laser-alloyed at velocities higher than 25 m/min, and gradually decreases going deeper into the laser remelted zone.
Answer 22
Thank you for this remark. Additional information has been added to the manuscript. We did not defined proportions of particular phases because the amount of Cu and Ni from substrate was equal for each coating. It can be assumed that there is more nickel borides than Cu-B eutectic in coatings because nickel is the main element of the substrate. Thank you for this valuable remark. We will include information about phase composition in coatings in our further publication.
Comment 23
Line 303. Authors state that “remelting Monel 400 with diode laser (by the way, it’s laser diode, and not diode laser) leads to obtain dendritic structure in which orientation is perpendicular to the surface”. However, on the basis of the discussion of results at lines 170-187, this is only partially true. Authors indeed write (line 175) “Different orientations in remelted areas result from complicated system of thermal gradients occurring in laser track during crystallization”. Authors are invited to write conclusions more in line with the discussion of results.
Answer 23
Thank you for this remark. According to the suggestion, revisions have been made in the manuscript.
The authors can’t agree with "(by the way, it's laser diode, and not diode laser)" because we have names: argon laser, gas laser etc. so also diode laser.
Round 2
Reviewer 1 Report
I hope the authors will carried out additional experiments of the structure and chemical bonding of the alloyed parts in the samples in the nearest future.
Reviewer 3 Report
I'm happy with the revised version of the manuscript.
The paper is now more readable, errors have been corrected, and missing info have been added properly.
I think that the paper is now ready to be published.